

# All that is gold does not glitter? Age, taxonomy, and ancient plant DNA quality

JinHee Choi, HyeJi Lee and Alexey Shipunov

Department of Biology, Minot State University, Minot, ND, USA

## ABSTRACT

More than 600 herbarium samples from four distantly related groups of flowering plants were used for DNA extraction and subsequent measurements of DNA purity and concentration. We did not find any significant relation between DNA purity and the age of the sample. However, DNA yields were different between plant groups studied. We believe that there there should be no reservations about "old" samples if the goal is to extract more DNA of better purity. We argue that the older herbarium samples are the mine for the future DNA studies, and have the value not less than the "fresh" specimens.

## INTRODUCTION

Numerous museums and agencies collect millions of herbarium specimens, establishing "dry gardens" that preserve plant species diversity. To date, many of the studies related with the herbaria are morphological. However, there are a growing number of investigations where extracting of DNA from the dried samples (*Savolainen et al., 1995*) is done in order to investigate the evolution of plants.

The study of "ancient DNA" originated from herbarium collections, even from old ones, and has already provided many useful results which argue strongly for the preservation of museum collections. DNA from herbarium samples was used to find origins of the European potato from samples before 1850 (*Ames & Spooner, 2008*). DNA was successfully extracted, amplified, and sequenced from two historical *Pinus* collections dated from 1811 (*De Castro & Menale, 2004*). Roullier and others (*2013*) used samples dated back to 1769 to clarify the distribution routes of sweet potato in the Oceania. The recently published research on the origin of watermelon (*Chomicki & Renner, 2015*) used the successfully amplified and sequenced DNA from the herbarium sample of watermelon holotype collected in 1773.

Multiple improvements for the herbarium extraction protocols were invented during the last decades (*Drábková, Kirschner & Vlĉek, 2002*; *Costa & Roberts, 2014*; *Drábková, 2014*). Also, the importance of understanding of how exactly DNA quality and herbarium sampling are related (*Erkens et al., 2008*; *Adams & Sharma, 2010*; *Staats et al., 2011*; *Särkinen et al., 2012*; *Neubig et al., 2014*) is apparent. It is clear that size of the DNA fragments is negatively related with age (*Erkens et al., 2008*; *Adams & Sharma, 2010*; *Neubig et al., 2014*). However, not all questions are answered here. For example, is there dependence between the spectrophotometric purity and concentration of the extracted

Corresponding author
Alexey Shipunov,
alexey.shipunov@minotstateu.edu

DNA and the age of sample? Is the quality of the extracted DNA generally different between taxonomic groups of plants? To answer, these questions require the significant amount of samples and the reasonable uniformity of DNA extraction conditions (*Erkens et al., 2008*; *Neubig et al., 2014*).

In our laboratory, we are currently running several taxonomic projects that necessitate the extraction of DNA from herbarium samples of multiple different lineages of plants and ages of material, namely (1) Plantagineae tribe of Plantaginaceae (*Aragoa*, *Littorella* and *Plantago* s.l.), (2) suborder Buxinae (or order Buxales: *Didymeles* from Didymelaceae and all genera of Buxaceae, including the newly placed *Haptanthus*), (3) Picramniaceae (*Picramnia*, *Alvaradoa* and recently described *Nothotalisia*), and finally (4) genus *Aronia* from Rosaceae. These four plant lineages are dramatically different in their evolutionary origins, geography, and ecology; these differences increase our chances to reveal the reliable patterns in the results of DNA extraction. Also, we accumulated the amount of data that allows for the comprehensive statistical analysis and comparisons. Consequently, as a secondary effort, we have taken this opportunity to compare the DNA extraction results, which in turn might help to answer questions raised above.

## MATERIALS AND METHODS

Overall, we used standard approaches to the DNA extraction and employed commercial DNA extraction kits. DNA was extracted using mostly a MO BIO PowerPlant DNA Isolation Kit (MO BIO Laboratories, Carlsbad, California, USA) and NUCLEOSPIN Plant II Kit (MACHEREY-NAGEL GmbH & Co. KG, Düren, Germany). Dry plant leaf material (typically, 0.05–0.09 g) was powdered using a sterile mortar and pestle and then generally processed in accordance with the supplied protocol. However, based on the results from *Costa & Roberts (2014)* and our own experience, we increased the lysis time to 30 min and used thermomixer on the slow rotation speed (350 rpm) instead of water bath. Nanodrop 1000 Spectrophotometer (Thermo Scientific, Wilmington, Delaware, USA) was used to assess the concentration and purity (the 260/280 nm ratio of absorbance) of DNA sample.

DNA concentration and purity values revealed with Nanodrop 1000 Spectrophotometer are likely not ideal tools to characterize the ultimate quality of the extracted DNA. However, in our 628 samples they worked as reliable and statistically significant predictors of the amplification and sequencing success (data not shown, but in a forthcoming complementary manuscript). Spectrophotometric DNA purity is especially useful for this purpose. Consequently, we regard them as useful tools which allow us to assess the DNA quality.

At the time of this manuscript, we extracted DNA from 628 samples: 386 Plantagineae, 73 Buxineae, 134 Picramniaceae and 35 *Aronia*, respectively. The age of our samples vary widely (Fig. 1) from recently collected to more than 150 years old, but most samples were collected 30–70 years ago. Herbarium tissue samples were obtained from the largest herbaria in USA (US, NY, HUH, MO, CAS, UC, JEPS, F) and some of them from Russian herbaria (LE, MW, MHA) with the kind permission of the herbarium curators. All samples were photographed to provide vouchers for the DNA isolation from each specimen.
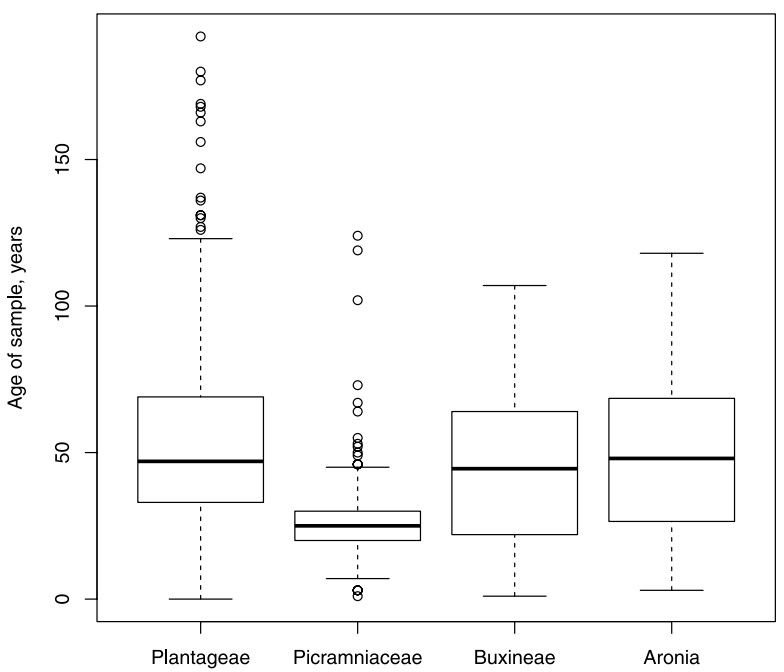

**Figure 1 Box plots showing the distribution, mean and variability of the ages of our samples.** The freshest samples were 3–5 years ago while the oldest was collected in 1820.

For all statistic calculations, R environment (*R Core Team, 2014*) was used.

## RESULTS

Four studied groups of plants are not closely related. In fact, they dramatically differ in ecology and geography in morphology. Consequently, in order to better understand the results of this study, we analyzed the data in subsets of the different included plant groups and their comparisons between DNA concentration and purity and age are as follows:

(1) Plantagineae. The statistical analysis of the available data revealed that there was no relation (Fig. 2A) between concentration of the DNA and age of sample (adjusted R-squared of the linear model $= -0.001$, $t$-test $p$-value for the slope $= 0.229$). There was also no significant relation between the purity of the DNA, measured from the absorption ratio 260/280 nm, and age of sample (adjusted R-squared of linear model $= 0.004$, $t$-test $p$-value for the slope $= 0.106$). Naturally, there was a positive (but still weak) relation (Fig. 2B) between the concentration of the DNA and sample dry-weight prior to extraction (adjusted R-squared of linear model $= 0.042$, $t$-test $p$-value for the slope $\ll 0.001$).

(2) Buxineae. Similarly to Plantagineae, Buxineae had no (or only weak) relation between age of the sample and the DNA purity (adjusted R-squared of linear model $= 0.031$, $t$-test $p$-value for the slope $= 0.079$). Concentration of the DNA and the age of the sample were also unrelated (adjusted R-squared of linear model $= -0.012$, $t$-test $p$-value for the slope $= 0.723$).
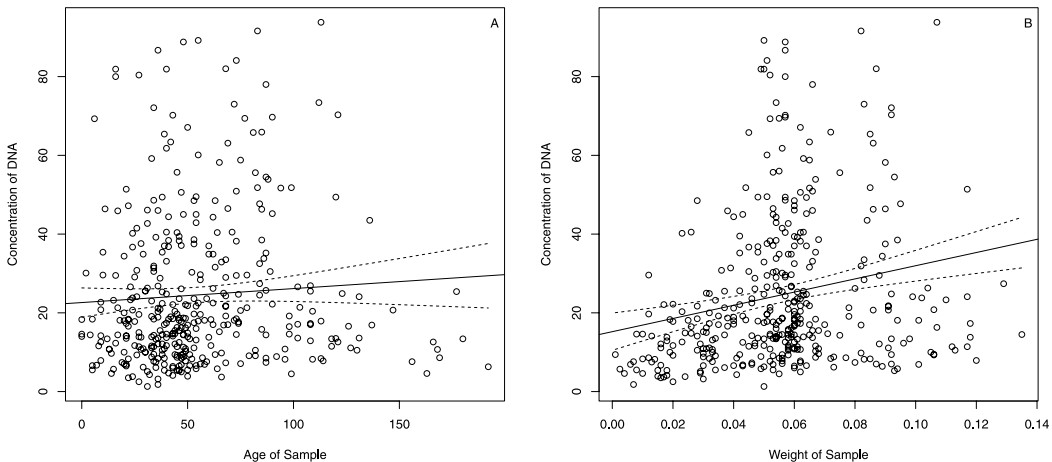

**Figure 2** **DNA concentration vs. sample age (A) and sample weight (B) for the Plantageae samples in this study.** The dashed lines indicate 95% confidence bands.

(3) Picramniaceae. The linear model was not statistically significant (adjusted R-squared of linear model $= -0.001$, $t$-test $p$-value for the slope $= 0.34$). The same was true for the concentration vs. age (adjusted R-squared of linear model $= 0.005$, $t$-test $p$-value for the slope $= 0.199$).

(4) *Aronia*. The discussed parameters were not statistically related as in the previous three cases. For purity vs. age, adjusted R-squared of linear model $= 0.035$, $t$-test $p$-value for the slope was equal to 0.145. For concentration vs. age, adjusted R-squared of linear model $= 0.026$ and $t$-test $p$-value for the slope $= 0.176$.

## All four groups

The average purity of DNA was different between our plant groups: while most of them are around 1.7 260/280 (1.7, 1.67 and 1.72 for Plantagineae, Buxineae and *Aronia*, respectively), Picramniaceae yields DNA of 1.57 average purity. This is supported with Kruskal–Wallis test ($p$-value $\ll 0.05$). Pairwise comparison of groups (Wilcoxon tests with Holm correction) revealed that the difference attributed to Picraminaceae was significant ($p$-value $\ll 0.05$) or marginally significant ($p$-value $= 0.06$) with an exception of *Aronia* ($p$-value $= 0.20$). Average ages of our samples (Fig. 1) were also different by plant group but in that case Picraminaceae samples were significantly younger (Kruskal–Wallis test $p$-value $\ll 0.05$).

Since concentration of DNA was related with sample weight (Fig. 2B), we compared the normalized concentration (concentration divided by weight) between four groups. This analysis revealed some differences: 386.4, 445.0, 563.2 and 618.3 for Plantagineae, Picramniaceae, Buxineae and *Aronia*, respectively. However, while the Kruskal–Wallis test was significant ($p$-value $\approx 0.00$), pairwise tests (Wilcoxon tests with Holm correction) revealed that only two differences were statistically significant—Plantagineae from Buxineae and from *Aronia* ($p$-values $< 0.05$).

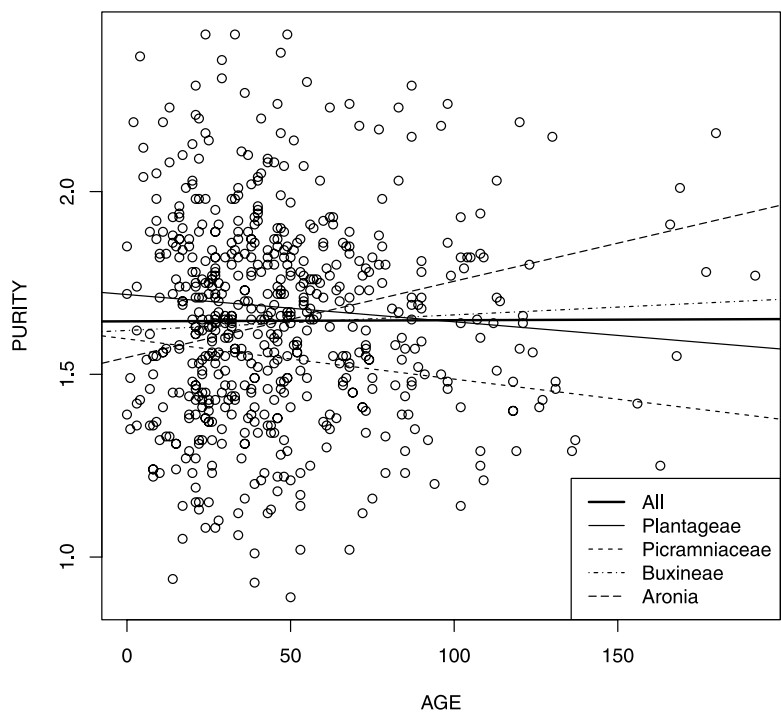

**Figure 3 DNA purity *vs.* sample age for all four groups studied.**

The comparative analysis of four linear models representing dependence of DNA purity from the relative age of sample (Fig. 3) revealed that while all intercepts were statistically different or marginally statistically different (*p*-values vary from 0.065 for *Aronia* to 0.007–0.008 for other groups), slopes were not different with the exception of Buxineae (*p*-value = 0.022). Moreover, AIC value was more optimal for the model without slopes (1535.7 vs. 1537.6). The overall significance of last model was reliable (Fisher test = 3.553, $df = 4$ and 611, *p*-value $\approx 0.01$).

### DNA extraction kits

The comparison of two age/purity linear models, each represented one protocol (142 and 244 extractions from different samples) found that while there was a slight difference in purity (Wilcoxon test *p*-value = 0.015), there was no significant differences between intersects and slopes of two models (all *p*-values > 0.05).

## DISCUSSION AND CONCLUSIONS

As all individual models suggest, there is no relation between age of sample and DNA purity. Consequently, there should be no reservations about "old" samples if the goal is to extract more DNA of better purity. Significant attention was paid to the DNA extraction protocols (*Drábková, Kirschner & Vlček, 2002*; *Costa & Roberts, 2014*; *Drábková, 2014*), and we believe that for the samples originated from herbarium collections the more attention should be devoted to the methods used on the subsequent stages of DNA processing (*Telle & Thines, 2008*; *Kuzmina & Ivanova, 2011*; *Samarakoon, Wang & Alford, 2013*). It is clearly

shown that the size of the extracted DNA fragments decreases with age (*Erkens et al., 2008*; *Adams & Sharma, 2010*; *Neubig et al., 2014*). Nevertheless, the fragment size is not the ultimate predictor of the sequencing success. Smaller ("barcoding") DNA markers (*Kuzmina & Ivanova, 2011*) as well as next-gen sequencing techniques (*Staats et al., 2011*) decrease the importance of this relation.

Another interesting result is that differences between studied taxonomical groups exist but were restricted with purity and relative concentration of samples. Interaction between age and purity (slopes of trend lines in Fig. 3) is not generally significant and therefore is not constrained taxonomically. This proves our hypothesis that non-significance of purity/age relation is the general phenomenon. Also, two commercial protocols used did not influence this dependence.

While the observed features of *Aronia* (non-significance of many comparisons and visually positive purity/age slope) are probably results of under-sampling (35 samples only), the significantly lower quality of Picramniaceae DNA begs explanation. Most likely, it is due to the complicated collection conditions (*Adams, 2011*): most species grow in wet tropical forests (*Thomas, 2004*) and therefore drying Picramniaceae samples is generally more complicated and at least longer than samples of plants from three other groups. Treatment for the pest control could be another important factor which lowered the DNA quality of the imported herbarium. There is also an evidence of the multiple secondary compounds specific for Picramniaceae (*Jacobs, 2003*). These compounds might also influence the final condition of the extracted DNA (*Särkinen et al., 2012*). Similar constraints were found by *Neubig et al. (2014)* in their investigation of DNA extracted from the variety of plant groups where some families (e.g., Melastomataceae) showed significantly lower extraction success.

In all, we believe that our research emphasizes the importance of museum collections of any age. We argue that the older herbarium samples are the mine for the future DNA studies, and have the value not less than the "fresh" specimens.

## ACKNOWLEDGEMENTS

We are grateful to the curators of US, NY, HUH, MO, CAS, UC, JEPS, LE, MW and MHA herbarium collections for the permission to work with their herbarium material. We thank our colleagues from the Barcoding of Life Consortium (especially Maria Kuzmina) for the fruitful collaboration for the SAPNA project, and our reviewers for their help in improving this manuscript.

### Funding

We received financial support from the College of Art and Sciences and the Department of Biology of Minot State University for the collection trips. From May 2014, this research is supported by North Dakota INBRE. The funders had no role in study design, data collection and analysis, decision to publish, or preparation of the manuscript.

## Grant Disclosures

The following grant information was disclosed by the authors:

College of Art and Sciences.

Department of Biology of Minot State University.

North Dakota INBRE.

## Competing Interests

The authors declare there are no competing interests.

## Author Contributions

- JinHee Choi performed the experiments, contributed reagents/materials/analysis tools, wrote the paper, reviewed drafts of the paper.
- HyeJi Lee performed the experiments, analyzed the data, contributed reagents/materials/analysis tools, wrote the paper, reviewed drafts of the paper.
- Alexey Shipunov conceived and designed the experiments, analyzed the data, contributed reagents/materials/analysis tools, wrote the paper, prepared figures and/or tables, reviewed drafts of the paper.

## Data Deposition

The following information was supplied regarding the deposition of related data:

Data can be downloaded from: http://ashipunov.info/shipunov/open/nanodrop.zip.

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
