# Peer review of "All that is gold does not glitter? Age, taxonomy, and ancient plant DNA quality"

_PeerJ, doi:10.7717/peerj.1087_

## Round 0.1 · original submission · Major Revisions

As the reviewer comments indicated, all recognize the importance of your work relative to the potential importance of herbarium material, and all offer suggestions for editorial changes to improve the clarity of your message. The most substantive question raised by the reviews is that of Reviewer 2, who argues that more details must be provided on the quality of the DNA to properly address the intent of your paper. In making your revision/rebuttal letter this is a key issue I would like you to address.

Reviewer 1 ·

Basic reporting

The article is written in an understandable manner, however, grammatical errors exist. I have noted this on the Word-track changes document attached to my review. There are some significant short comings in the introduction and discussion of the research shown and presented here. These are also noted in detail in my attachment. I believe they can be addressed in a revision. The choice of structure of the results section allows for too many repetitive sentences and would be greatly improved by a Table and the combination of Figures 2 and 3. Additionally, the results section does not need to be divided in the sections that the authors chose to divide them. Detailed comments are on my attachment.

Experimental design

It should be made clear that the authors are presenting an artifact of doing a different taxonomic research project all together. This is of course acceptable, but it is important that it is noted that these results are presented as a side-product of a larger experiment.

Validity of the findings

I have strong opinions on basing such strong conclusions on the data based on the use of the NanoDrop spectrophotometer for DNA concentration and purity for this study. Please see the comments within my attached document.

However, I do believe that given what they had to work with, their conclusions are correct. The authors need to admit the short comings of the instrumentation used.

Additional comments

This data illustrates a partial story of the larger problem of creating modern uses of dried plant material housed in Herbaria. I am very happy to see such a large dataset. I hope, if possible, the authors will be able to later report on the ability of the DNA to be amplified with PCR from these samples. Then the results would be much more important.

Annotated reviews are not available for download in order to protect the identity of reviewers who chose to remain anonymous.

·

Basic reporting

The premise of this study was to assess DNA “quality” of herbarium specimens. The writing is clear, but is lacking in a lot of background information with regards to DNA quality.

Experimental design

This is not a measure of DNA quality because it provides no data on the length and intactness of DNA in samples. In other words, the data are not particularly meaningful with regards to DNA and the conclusions are inappropriate given the data.

Validity of the findings

Other papers have tried to address DNA quality, neither of which are cited here, nor many of the important papers cited in the attached papers!

Additional comments

This paper would need significant changes to reword regarding what is being measured and analyzed as it is very misleading. If you would like some more references regarding DNA quality in plants, please let me know: Kurt Neubig ([email protected])

·

Basic reporting

This article assesses DNA concentration by age of herbarium plant samples. The authors show convincingly that old herbarium samples preserve DNA sufficiently well for analysis of plant evolution. This is important because there is pressure in some areas to reduce museum holdings. This paper could argue that herbariums should preserve samples, even old samples, as a mine for future DNA studies. If I were the authors, I would make this point explicit.


The review section in the introduction is excellent. This is very useful.
The materials and methods are succinct and sufficient.

Experimental design

The submission accomplishes all of its goals.
The submission describes original primary research.
The submission clearly defines the research question.
HOWEVER, the relevance is understated by the authors. This report could argue strongly for the preservation of museum specimens.
The investigation was conducted rigorously to a high technical standard.
Methods are reproducible.

Validity of the findings

The data are robust and statistically sound. They are presented graphically by the authors. There are several grammatical errors that muddy the conclusions.

Additional comments

I am confused by this sentence for the Picramniaceae results and I suggest that the authors rewrite it.
“While the visual inspection of the purity vs. age relation might leave an impression that in Picramniaceae, results were corresponded with the hypothesis of age-related DNA degradation, the linear model was not statistically significant . . .”

The first paragraph of Discussion and conclusions confuses me and should be rewritten.

Please define “taxonomical constraints”.

Change “the significantly lower quality of Picramniaceae DNA needs the separate explanation” to “the significantly lower quality of Picramniaceae DNA begs explanation.”

“no need to be afraid” change to “there should be no reservations”
“more attention should be paid on the PCR” change to “more attention should be paid to PCR methods”

Figure 1 Y axis needs a label

I was distracted by the title - All that is gold does not glitter. I spent 20 minutes contemplating how a phrase from Tolkien (and previously Shakespeare and Chaucer) related to this work. I think the authors mean that herbaria seem mundane, but are a treasure of DNA data. If I understand the literary reference correctly, then I suggest that the title is fine. If I came to the wrong conclusion, then perhaps this phrase should be deleted from the title.

---

## Round 0.2 · Minor Revisions

The manuscript will be acceptable for publication with the minor changes as indicated by reviewer 1 (mostly this is just the addition of Figure 3 to agree with the text),

Reviewer 1 ·

Basic reporting

The only missing piece to the revision is the clarification of the addition of Figure 3, the comparison between leaf tissue weight and nanodrop success. This figure was not included in the revision, but was written that it was added. This changes the final figure to be Figure 4.

There a few more small changes that are highlighted in the attached revision document.

Experimental design

No changes recommended. The authors corrected all aspects that were requested.

Validity of the findings

Again. Same as above.

Additional comments

Please see the comments on the revision attachment.

Annotated reviews are not available for download in order to protect the identity of reviewers who chose to remain anonymous.

·

Basic reporting

This article is now ready for reporting.

Experimental design

The experimental design is sound.

Validity of the findings

The authors adjusted their findings following reviewers comments. This part is fine now.

Additional comments

The paper is sound and ready to publish.

---

## Round 0.3 · accepted · Accept

Thanks for your hard work on the revisions, and careful attention to comments.